# Frequency and Time Domain Analysis of EEG Based Auditory Evoked Potentials to Detect Binaural Hearing in Noise

**DOI:** 10.3390/jcm12134487

**Published:** 2023-07-04

**Authors:** Eva Ignatious, Sami Azam, Mirjam Jonkman, Friso De Boer

**Affiliations:** College of Engineering and IT, Charles Darwin University, Casuarina 0810, Australia; eva.ignatious@cdu.edu.au (E.I.); mirjam.jonkman@cdu.edu.au (M.J.); friso.deboer@cdu.edu.au (F.D.B.)

**Keywords:** auditory evoked potential (AEP), electroencephalogram (EEG), homophasic, antiphasic, time domain analysis, frequency domain analysis

## Abstract

Hearing loss is a prevalent health issue that affects individuals worldwide. Binaural hearing refers to the ability to integrate information received simultaneously from both ears, allowing individuals to identify, locate, and separate sound sources. Auditory evoked potentials (AEPs) refer to the electrical responses that are generated within any part of the auditory system in response to auditory stimuli presented externally. Electroencephalography (EEG) is a non-invasive technology used for the monitoring of AEPs. This research aims to investigate the use of audiometric EEGs as an objective method to detect specific features of binaural hearing with frequency and time domain analysis techniques. Thirty-five subjects with normal hearing and a mean age of 27.35 participated in the research. The stimuli used in the current study were designed to investigate the impact of binaural phase shifts of the auditory stimuli in the presence of noise. The frequency domain and time domain analyses provided statistically significant and promising novel findings. The study utilized Blackman windowed 18 ms and 48 ms pure tones as stimuli, embedded in noise maskers, of frequencies 125 Hz, 250 Hz, 500 Hz, 750 Hz, 1000 Hz in homophasic (the same phase in both ears) and antiphasic (180-degree phase difference between the two ears) conditions. The study focuses on the effect of phase reversal of auditory stimuli in noise of the middle latency response (MLR) and late latency response (LLR) regions of the AEPs. The frequency domain analysis revealed a significant difference in the frequency bands of 20 to 25 Hz and 25 to 30 Hz when elicited by antiphasic and homophasic stimuli of 500 Hz for MLRs and 500 Hz and 250 Hz for LLRs. The time domain analysis identified the Na peak of the MLR for 500 Hz, the N1 peak of the LLR for 500 Hz stimuli and the P300 peak of the LLR for 250 Hz as significant potential markers in detecting binaural processing in the brain.

## 1. Background

Hearing loss is a prevalent health issue that affects individuals from various cultures, races, and age groups. The negative impact on a person’s quality of life can be considerable, particularly if the condition goes undiagnosed [1]. For children, undetected hearing loss can impede development, and early identification and intervention are critical for them to acquire essential skills at a similar pace to their peers. Thus, it is imperative to detect any form of hearing loss as early as possible [2]. Research has revealed that hearing loss can range from mild to profound, and that a broad spectrum of contributing factors exists [3,4,5,6].

Hearing loss can be categorized as either sensorineural or conductive, with sensorineural hearing loss affecting the inner ear and nervous system, while conductive hearing loss can be caused by malformations or diseases of the outer ear, the ear canal, or the middle ear structure. Both sensorineural and conductive hearing loss can interfere with cognitive development, interfering with auditory processing and binaural hearing development [7,8]. Binaural hearing is essential for sound localization, sound segregation, and understanding speech in noisy environments. Any dysfunction in one or both ears can disrupt the mechanism of binaural processing, resulting in difficulties in sound perception [9].

Binaural hearing is the ability to integrate information received simultaneously from both ears, enabling individuals to identify, locate, and separate sound sources. This ability is crucial for understanding speech in noisy environments, known as the “cocktail party effect” [10]. Any dysfunction in one or both ears, including unilateral hearing loss (UHL) and bilateral hearing loss (BHL) can disrupt the binaural processing mechanisms [11]. Previous research has shown that untreated binaural hearing impairment can significantly impede a child’s development, including verbal cognition skills [12]. Binaural hearing and other aspects of auditory functioning develop before adolescence. Adults can suffer from binaural hearing impediments, which can affect their quality of life and limit further cognitive development. As age increases, neural synchrony in the central auditory system deteriorates, contributing to the difficulty in perceiving temporal cues of sound for older people [13,14].

Auditory processing begins in the brainstem and mesencephalon, where responses such as auditory reflexes are coordinated, followed by processing in the auditory cortex of the temporal lobe [15]. Binaural hearing processing in the auditory cortex contributes to sound localization, differentiation, and delineation of important auditory sources and noise. Binaural hearing and other aspects of auditory functioning mature throughout adolescence and require auditory stimuli for full development. Without this input, auditory functioning, and therefore cognitive development, is at risk [16].

Testing for hearing impediments is diverse and well documented. While psychometric methods are commonly used, measures such as electroencephalograms (EEGs) have proven their ability to detect hearing disorders in infants [17]. The EEG method eliminates reliance on participant literacy and communication skills and is relatively easy to perform [18], making it highly suitable for young children [19]. Although collecting an EEG can be more challenging than a traditional hearing assessment, using the correct stimulus parameters, recording standards, and processing techniques can make this method suitable and feasible for binaural hearing assessment [20,21].

Auditory evoked potentials (AEPs) refer to the electrical potentials from any part of the auditory system, from the cochlea to the cerebral cortex, which are evoked by externally presented auditory stimuli. AEPs can be used to assess neurological integrity and auditory function [22]. Auditory evoked potentials (AEPs) are an essential tool for assessing auditory function and diagnosing hearing disorders [23,24]. AEPs are non-invasive and can provide valuable information about the function and integrity of the auditory system [24]. AEPs are categorized based on the time interval between the onset of the auditory stimulus and the peak of the evoked response [25]. The major AEP components are the auditory brainstem response (ABR), the middle latency response (MLR), and the late latency response (LLR) [26].

The ABR is the earliest AEP component, occurring within the first 10 ms after the presentation of an auditory stimulus. It reflects the activity of the auditory nerve and the brainstem and is commonly used to assess hearing sensitivity and diagnose hearing disorders [27]. The MLR is a later AEP component, occurring between 10 and 80 ms after the presentation of an auditory stimulus [28,29]. It reflects the activity of the auditory cortex and is thought to be involved in the processing of complex auditory stimuli, including speech. A recent study [30] investigated the use of the MLR to evaluate auditory processing in children with autism spectrum disorders (ASD). They found that the MLR was reduced in children with ASD compared to typically developing children, suggesting that the MLR can be a useful tool for identifying auditory processing deficits in individuals with ASD. The LLR is the latest AEP component, occurring between 50 and 500 ms after the presentation of an auditory stimulus. It reflects the activity of higher-order auditory processing areas, including the temporal and frontal lobes, and is thought to be involved in cognitive and attentional processing of auditory stimuli. In recent years the use of LLRs to investigate various hearing disorders has been explored [31,32]. AEPs have also been used to investigate the effects of different types of noise on auditory processing and to evaluate the neural mechanisms underlying sensory integration. Overall, AEPs are a valuable tool for evaluating auditory function and diagnosing hearing disorders [33].

The ABR, MLR, and LLR are the major AEP components, each reflecting different parts of the auditory pathway’s electrical activity. AEPs have the potential to provide important information for the management and treatment of hearing disorders, as well as to advance our understanding of the neural mechanisms underlying auditory perception and cognition [34,35,36]. Further research is needed to fully understand the neural mechanisms underlying AEPs and their clinical and scientific implications.

The use of auditory evoked potentials (AEPs) in investigating binaural hearing has led to the identification of MLR and LLR as two main components of interest. Research related to the MLR has primarily focused on its sensitivity to binaural processing. Studies have shown that the MLR is modulated by differences in interaural time and intensity cues, which are important for sound localization and binaural fusion [29]. Furthermore, the MLR has been shown to be associated with the perceptual grouping of sounds, such as the segregation of speech from background noise [37,38]. However, the literature on the MLR in binaural hearing is limited, and more research is required to comprehend its role in binaural hearing. The LLR has also been studied in the context of binaural hearing [39]. Meanwhile, studies have shown that the LLR is modulated by binaural cues, including interaural time and intensity differences, and is sensitive to the spatial location of sounds [40], suggesting its potential role in auditory scene analysis through its sensitivity to changes in binaural cues over time [41]. However, the literature on the LLR in binaural hearing is still limited, demanding further research to fully understand its contribution to binaural processing.

The literature on the EEG and AEPs is diverse, but there is a lack of relevant literature regarding binaural hearing assessment and testing stimuli for AEP responses, indicating an avenue for future studies. A review of the literature reveals a gap in knowledge regarding the use of the EEG for binaural assessment. Specifically, research pertaining to the Auditory MLR and LLR indicates a need for further investigation and contribution to the existing body of knowledge. This study aims to fill this gap by exploring the potential of AEPs for binaural hearing assessment through MLR and LLR analysis.

ERP signals from the brain can be analysed in different ways which includes time domain analysis by examining AEPs, or the frequency domain analysis that uses methods such as FFT (Fast Fourier Transform) or Welch’s Periodogram (Pwelch) [42,43,44]. The Pwelch method, a spectral decomposition technique, calculates the Power Spectral Density (PSD) for EEG data, providing valuable information about the spectral content and power across different frequency bands. It reduces variance and allows for high accuracy and resolution in PSD estimation, making it useful in ERP data analysis with low signal-to-noise ratios [45]. The Pwelch method can handle signals with non-uniform sampling rates and offers insights into the underlying neural mechanisms of cognitive and sensory processing [44]. AEPs can be analysed in the time domain through averaging techniques, allowing for the examination of peak amplitudes, latencies, and interpeak latency differences under various conditions [46,47]. Overall, the AEP analysis and Pwelch method in both time and frequency domains are essential in understanding brain function and neurological disorders.

This study aims to investigate the use of audiometric EEGs as an objective method to detect binaural hearing. Limited research on the MLR and LLR with binaural hearing stimuli creates an opportunity for novel contributions to the existing knowledge. For this study, the BMLD (Binaural Masking Level Difference) test, which has been recommended for measuring binaural hearing loss, has been taken as a starting point. The stimuli used in the BMLD trials are employed in the current Audiometric EEG study to investigate the impact of binaural phase shifts of the auditory stimuli in the presence of noise.

## 2. Materials and Methods

This study was conducted at Charles Darwin University in Australia, and the experimental protocols were approved by the Human Ethics Committee of the University, as outlined in H18014—Detecting Binaural Processing in the Audiometric EEG. Prior to the experiment, written consent was obtained from all volunteers indicating their willingness to participate in the various hearing tests, including EEG measurements. A plain language statement (PLS) was provided outlining the details of the study and to provide an overview of the experimental process. A questionnaire was completed by each subject to ensure a healthy otological history. The experimental set up and hardware was the same as explained in [20].

### 2.1. Participants

The sample for the present study included thirty-five participants (23 males and 12 females) between the ages of 18 and 33 years old (mean age 27.36), with a hearing threshold value between 0 and 20 dB hearing level (HL) in both ears, as confirmed by pure-tone audiometry. Pure-tone audiometry was conducted initially to measure the hearing threshold levels of the participants at different frequencies, in accordance with the relevant Australian Standards, and to determine whether they were acceptable [20]. All the selected thirty-five subjects had normal audiograms with a hearing threshold value between 0 to 20 dB hearing level (HL) for the frequency range of 125 Hz to 8 KHz [20].

Participants were also required to read and understand the PLS. Additionally they were asked to complete a questionnaire to ensure no noticeable otological issues were detected in the past or present. Exclusion criteria included participants who were under 18, had severe hearing loss conditions or cognitive impairment, had a cochlear implant or other implantable hearing device, or had severe or debilitating tinnitus. Additionally, pregnant women were excluded due to potential risk to the foetus. The inclusion and exclusion criteria were carefully considered to ensure that the trial results were valid and could be applied to the target population. The subjects were prepared for the experimental process as explained in the preliminary study conducted [20].

### 2.2. Auditory Stimuli

The stimuli used in the study were Blackmann windowed pure tones of different frequencies: 125 Hz, 250 Hz, 500 Hz, 750 Hz and 1000 Hz. Blackman windowing was performed to ensure a smooth acoustic transition at the start or the end of the stimulus and thus to reduce spectral splatter of the signal. The tones were embedded in a 10 Hz bandwidth Gaussian noise masker throughout. The centre frequency of the masker and the frequency of the tone were the same for these trials. The stimuli were 518 ms and 548 ms in duration, generated in MATLAB R2017b with a signal of 18 ms and 48 ms, respectively. The durations were chosen to correspond to the duration of signals that can be used for the generation of AEPs: 18 ms for the Middle Latency Response (MLR) and 48 ms for the Late Latency Response (LLR). The level for the masker was set at 20 dB whereas the tone was at 40 dB. The sampling frequency was set to 19.2 kHz. The generated stimuli of 18 ms and 48 ms, for the MLR and LLR respectively, are illustrated in Figure 1 and Figure 2. The stimuli were presented in a predefined sequence: blocks of 25 homophasic stimuli followed by 25 antiphasic stimuli. A total of 1000 trials were carried out per subject, resulting in the generation of 500 antiphasic and 500 homophasic ERPs for each subject. The total time for 1000 trials was 8.633 min and 9.133 min for the 18 ms and 48 ms stimuli, respectively. On average, healthy adults have an attention span in the range of 10 to 20 min, so a short trial reduces the risk of hearing fatigue and adaptation during the experimental trial [48]. Subjects were asked to relax before and between the experiments, in order to ensure quality data acquisition. The stimuli were delivered to the ER.2 insert earphones via an external sound card (Creative Sound Blaster Omni Surround 5.1) at a 60 dB sound pressure level. The pure tone stimuli consisted of a sinusoidal signal, So, and its opposite signal, Spi [20].

### 2.3. Electrode Placement

Electrical activity in response to auditory stimuli was recorded from 12 electrode sites on the scalp. The arrangement of specific electrode sites used for the recording of auditory evoked potentials (AEP) is shown in Figure 3 [49,50]. The reference electrode was positioned on the left earlobe, while the ground electrode was placed in the lower position on the forehead (FPz), following the 10–20 electrode placement system [51,52].

## 3. Results and Analysis

### 3.1. Signal Processing and Analysis

After the data acquisition process was completed, data processing and analysis were carried out offline. Figure 4 illustrates the workflow followed for data processing and analysis. The data were imported as an array into MATLAB. This contained the data of the captured EEG channels, the trigger channels and the stimulus channels. Further processing of signals was carried out in MATLAB R2017b using the EEGLAB v2019.1 toolbox. The sampling rate for EEG data acquisition is 19.2 kHz and the Nyquist frequency is 9.6 kHz, much higher than required for analysing the frequency ranges in the human AEPs. In the pre-processing stage, the EEG data were down sampled to 2048 Hz [53]. The process of down sampling ensures that the filtering process in the preprocessing stage is computationally efficient and improves the roll-off ability of the filters. The next step in data pre-processing was to extract the accurate trigger times by removing false triggers and checking whether 1000 triggers were found in total. The detection of the triggers was carried out by checking the time between two triggers (0.518 s or 0.548 s). If a shorter time delay was found, the corresponding trigger was rejected. The trigger channel data captured by the amplifier were used to synchronize the averaging process. The 18 ms input stimuli, masked in noise in both the homophasic and the antiphasic conditions, were used to evoke the Middle Latency Response (MLR) and the 48 ms input stimuli, masked in noise in both the homophasic and the antiphasic conditions, were used to evoke the Late Latency Response (LLR). As shown in Figure 5, the actual triggers were detected by analysing the right ear stimuli (Stim-R) and left ear stimuli (Stim-L) together with the trigger timing. Figure 5a,b, shows that the triggers at the start of each stimulus were detected accurately. Once the triggers were detected, artefacts and noise were removed from the EEG signals to obtain clear evoked responses for further processing and analysis [54]. The down sampled data were then filtered using an FIR filter with a Hamming window. A low cut-off frequency of 1 Hz was applied to the data to remove slow drift noise and DC components from the signal. The next stage of data processing involves epoch generation. The start of the epoch was determined using the trigger signal, which was delivered synchronously with the stimulus [20]. The duration of the epoch is based on the duration of the signal and the interval between the signals. Using the trigger signal, the responses to homophasic and antiphasic stimuli were identified. The trial was then cut into suitable time frames for analysis. Thus, the evoked potentials were then split in epochs (pre-defined short duration of time) [55]. In the present study, the time windows of interest are the MLR, which ranges from 20 ms to 100 ms, and the LLR from 50 ms to 500 ms after the start of the input stimuli. Epochs with amplitudes larger than 150 mV were rejected. Baseline correction was conducted for the remaining trials [56].

The remaining epochs were analysed in the time domain and frequency domain. As an initial step for the time domain analysis, epoch averaging was carried out separately for each of the twelve electrode locations. The averaged AEP signals from the accepted trials were then further analysed in the time domain. Frequency domain analysis, however, was carried out initially as individual epochs and in the later stage the averaging was carried out.

### 3.2. Frequency Domain Analysis

Frequency domain analysis is a widely used method for EEG analysis, and it is often conducted as the first analysis since it is a general and the most common method to understand the EEG data as a whole [57,58]. Frequency domain analysis is regarded as the most powerful and standard method for EEG analysis, compared to other methods [57]. It gives insight into information contained in the frequency domain of EEG waveforms by adopting statistical and Fourier transform methods [57]. Power spectral analysis is the most used spectral method since the power spectrum reflects the ‘frequency content’ of the signal or the distribution of signal power over frequency [57]. The frequency-domain feature analysis method mainly observes the frequency spectrum of the EEG signal of a certain length, which can obtain the distribution [59]. Hence, frequency domain analysis was used in the analysis of the electroencephalogram (EEG), particularly in the context of auditory responses. By transforming time-domain signals into the frequency domain, it may be possible to extract some information about the underlying neural processes that generate the signals in response to auditory stimuli [44,60]. In EEG signal analysis, frequency domain analysis typically involves the use of Fourier transform techniques such as the Fast Fourier Transform (FFT) or modified methods such as Welch’s periodogram (Pwelch) to estimate the spectral components of recorded EEG signals [45]. This technique provides valuable insights into the frequency components of EEG waveforms and is based on the mathematical principles of Fourier transforms and statistical analysis [42].

The Pwelch method is a common method of spectral decomposition of EEG signals used in ERP data analysis. The PSD calculated by the Pwelch method is computed by dividing the time signal into successive blocks, forming the periodogram for each block, and then averaging the periodogram results.

The epoched AEP data are then transformed into power spectral density (PSD) using Welch’s periodogram method using a Hanning window. The Welch periodogram is calculated using Equation (1).
(1)$$P_{xx\left(f\right)}=\frac{1}{N}{\left|\sum_{n=0}^{N-1}x\left(n\right)w\left(n\right)e^{-j2\pi {nfT}_{s}}\right|}^{2}\_\_\_\_\_\_\_\_\_\_\_\_\_\_\_\_$$
where the w(n) is the Hanning window function, defined as:$$w\left(n\right)=0.5\left[1-cos\left(\frac{2\pi n}{N-1}\right)\right]$$

Welch’s method was applied to the MLR and LLR epochs of the EEG signals for each electrode. In other words, Welch’s calculation was conducted on all the extracted epochs of MLR and LLR data for each channel and each subject. Welch’s method generated PSD values for every calculated frequency on the signal of each epoch. The PSDs from all trials were then averaged. Once the averaged PSD of every subject was calculated, further analysis was conducted on more specific frequency bands. In the current study, the frequency range of 15–50 Hz was the objective [20,50]. The PSDs of every channel in all subjects were then trimmed into 15–50 Hz. For further analysis, the trimmed frequency was then separated into 5 Hz bands, resulting in 7 bands: 15–20 Hz, 20–25 Hz, 25–30 Hz, 30–35 Hz, 35–40 Hz, 40–45 Hz, and 45–50 Hz. The PSD for every band was then averaged, with the high boundary of the band not included, i.e., for 15–20 Hz, the averaged PSD data does not include 20 Hz (15 ≤ f < 20). The results are seven mean PSD values for every channel and every subject for two input stimuli (homophase and antiphase) of five different frequencies. Figure 6a,b represent mean PSD values for every EEG channel for one subject study in the homophasic and antiphasic conditions, respectively.

The mean PSD values were then analysed using a statistical method to evaluate whether the PSD values were normally distributed. For this process, the Shapiro–Wilk test with alpha = 0.05 was employed to evaluate normality of the distribution among subjects for the same channel and frequency band. The results show that the data in general have *p* values less than 0.05, which means that the PSD value distribution among the subjects for each channel and frequency band is not normal. Figure 7 and Figure 8 show the MLR and LLR plots in response to the 500 Hz stimulus, for seven different bands for the Cz channel. Since parametric tests require normal distribution of data, a non-parametric test was chosen as the alternative for further evaluation. The Wilcoxon signed-rank test was chosen to evaluate the significance of the differences between responses to the antiphasic and homophasic stimulus conditions among subjects for the same channel and frequency band. The Wilcoxon signed-rank test uses two matched samples, comparing their rank within the population. It is known to be more robust for a small sample size, usually under 50 samples. The process uses an alpha value as an indicator to accept or reject the null hypothesis that there is no significant difference between the homophasic and the antiphasic conditions. The most common alpha value for the Wilcoxon test is 0.05, which is also used in this study. If the *p*-value is less than 0.05, the null hypothesis is rejected, which means that there is a significant difference between the homophasic and the antiphasic conditions. If the *p*-value is more than 0.05, the null hypothesis cannot be rejected, which means that there is no significant difference between the homophasic and the antiphasic condition. Table 1, Table 2, Table 3, Table 4, Table 5, Table 6, Table 7, Table 8, Table 9 and Table 10 show the Wilcoxon signed-rank significance test results for MLR and LLR data for five different frequencies for all twelve channels.

The results of the statistical analysis from Table 1, Table 2, Table 3, Table 4, Table 5, Table 6, Table 7, Table 8, Table 9 and Table 10 indicate that there exists a significant difference between antiphase and homophase PSD data among subjects for the midline channels of the MLR for 500 Hz stimuli. The difference is significant for the 20 Hz to 25 Hz and 25 Hz to 30 Hz frequency bands. The LLR shows significant differences between the homophasic and antiphasic conditions in various channels for the 500 Hz stimuli. In addition, for the 250 Hz stimuli, the midline and left channels recorded a noticeable difference between the two conditions. Most of the significant differences are in the 20 Hz to 25 Hz and 25 Hz to 30 Hz frequency bands.

The highlighted cells indicate that the *p*-values are less than alpha, which means that statistically significant differences exist in the PSD bands between the two conditions. Frequency band analysis is a useful tool for identifying the frequency range where the response to the binaural cue of phase reversal is most pronounced. The significant differences observed between the antiphasic and homophasic conditions support the notion that the interaural phase difference (IPD) is important for localizing sound sources. These findings suggest that the brain is particularly sensitive to IPD cues in certain frequency bands, which are used to determine the position of a sound source.

### 3.3. Time Domain Analysis

The time domain analysis aims to: (1) determine the effects of phase-shifted pure tone stimuli masked in noise of different frequencies on the MLRs and LLRs for normal hearing subjects; (2) compare the effects of ERP components with previously reported effects; and (3) determine which ERP components of MLR and LLR are dominant and contrast. The ERP data were processed using time-series domain analysis. The first step averaged the accepted epochs after epoch rejection in preprocessing. Next, the visual inspection was conducted to see whether there is a trend or pattern in the signal. This visual inspection was conducted by channel-wise and subject-wise plotting of the MLRs and LLRs, see Figure 9, Figure 10, Figure 11 and Figure 12. For the channel-wise plotting, data for every electrode from all subjects were gathered and grouped for the corresponding electrode. Each electrode group was then plotted and analysed visually. For subject-wise plotting, all electrodes for the same subject were plotted in the same Figure and analysed visually.

From visual inspection, it was found that the ERPs look similar among channels for each subject, as shown in Figure 11 and Figure 12. This finding is consistent with previous studies that have reported individual differences in topography, latency, and morphology of ERP components [61,62]. However, contradictory results are obtained when the ERPs are plotted for the same channel for all subjects, as shown by Figure 9 and Figure 10. This discrepancy suggests that there may be subject-specific differences in the ERPs [63,64]. For instance, one study [62] analysed ERP traces from 238 scalp channels averaged over 500 EEG epochs in a single subject and found that the ERPs were subject specific.

The next step was the extraction of peaks and peak-to-peak values of the MLR and LLR from every channel and subject. For the MLR, the ERP components to be extracted are Na, Pa, Nb, and Pb, while for the LLR, the ERP components are N1, P1, N2, P2 and P300. A representation of the peak components in the MLR and LLR waves is shown in Figure 13. The results were categorized into 10 MLR variables and 11 LLR variables for homophasic and antiphasic data as shown in Table 11. The mean and standard deviation values of all extracted MLR and LLR peaks are shown in Figure 14. Differences between the homophasic and antiphasic results were not immediately apparent. Nonetheless, some insight can be drawn from the results depicted in Figure 14.

Firstly, the Na peak of the MLR for the 500 Hz stimulus has the highest differences in mean values between the antiphasic and homophasic conditions with a low standard deviation. This suggests that the Na peak of the MLR for the 500 Hz stimulus may be sensitive to phase differences in the stimuli. Secondly, for the LLR, the highest average differences between the homophasic and the antiphasic conditions are found for the N1, N2, and P300 peaks for the 250 Hz stimuli and the N1 peak for the 500 Hz stimuli. This indicates that the N1, N2, and P300 peaks of the LLR for 250 Hz stimuli and the N1 peak for the 500 Hz stimuli may be more sensitive to phase differences. These findings suggest that the differences between the antiphasic and homophasic condition in mean values of ERP peaks vary across different frequencies of the stimuli and may depend on the type of AEPs. Previous studies on auditory middle latency responses (MLRs) have shown that the amplitudes of most MLR peaks increase and their latencies decrease with increasing stimulus intensity [65]. To evaluate the significance of ERP peak amplitudes for stimuli of different frequencies, a statistical test is required [61].

The Shapiro–Wilk test was conducted to examine the normality of the results. Since the visual inspection shows similarity across channels for the same subject, subject-wise data grouping for the tests was conducted, i.e., it was checked whether the distribution was normal among channels for the same subject. The normality results show that all peak categories have normal distribution for almost all subjects. Since the peak categories show a normal distribution trend, they may be averaged among subjects for each category separately. The results of averaged peaks among channels for each subject were converted into absolute values prior to the averaging of all subject peak values in each category. The use of absolute values when analysing peak amplitudes is consistent with the literature because it allows for the comparison of peak amplitudes across different conditions and subjects [66]. From the averaged results, the peaks of the MLR and LLR varied with stimuli of different frequencies.

It was then evaluated whether there is a significant difference in the MLR and LLR components between the homophasic and antiphasic for different stimulus frequencies. A two-sample *t*-Test was conducted for the mean peaks in the antiphasic and homophasic condition among subjects. The results show that there are significant differences in the Na and N1 peaks between the antiphasic and the homophasic condition for 500 Hz stimuli. In addition, the P300 peak of the LLR showed significant differences between the antiphasic and the homophasic condition for 250 Hz stimuli. The results are listed in Table 12 and Table 13.

## 4. Discussion

The findings of the current study suggest that for signals masked in noise, phase changes can have a significant effect on binaural processing in the human brain, as measured by auditory evoked potentials (AEPs). Our results showed that there were statistically significant differences between the AEP signals generated by antiphasic and homophasic stimuli, in both time and frequency domain features. The differences suggest that the brain can detect and process interaural phase differences, which may be important for spatial localization of sound sources and other aspects of auditory processing. The results are consistent with previous studies [67] that have shown that binaural stimulation results in larger cortical responses compared to monaural stimuli, and that the amplitude and latency of AEPs are dependent on the binaural difference. However, our study is unique in its focus on phase changes and its use of stimuli with frequency and noise parameters. The findings may improve our understanding of binaural processing in the human brain and lead to applications in the development of new objective hearing tests in the future. Our results indicate that the detection of phase differences may be an important factor in the “cocktail party” effect, whereby listeners are able to focus on a particular sound source in a noisy environment [68].

In the frequency domain study, we used the Pwelch method to calculate the power spectral density values of the MLR and LLR signals in various frequency bands to investigate the significance of phase differences in binaural processing [69]. Our results showed that the 20–25 Hz and 25–30 Hz frequency bands of the MLR and LLR signals had a significant difference for antiphasic and homophasic stimuli. These frequency bands correspond to the high beta and low gamma frequency range of the EEG [70]. The finding is consistent with previous research suggesting that sensory integration results in frequencies in the high beta and low gamma range, which may indicate conscious and accurate phase detection of auditory stimuli [70,71]. Further analysis revealed that the stimuli which resulted in statistically significant differences were 500 Hz for the MLR signals, and 250 Hz and 500 Hz for the LLR signals, mainly in the 20–25 Hz and 25–30 Hz frequency bands. These findings suggest that optimal binaural processing occurs at 500 Hz, in line with previous literature predicting that lower frequencies result in larger binaural masking level differences (BMLD) [72,73].

We also analysed the electrode locations that provided more significant differences between the homophasic and antiphasic condition. Our results show that the midline electrodes provided more significant differences in the MLR [74], while both the midline and left electrodes provided significant differences for the LLR signals [75]. This finding may indicate that midline electrodes could be more suitable to investigate the processing of pure tone stimuli in binaural hearing [76]. The left hemisphere of the brain, which is known to be important for processing the temporal aspects of sound, may also be involved in this processing [77]. It is also in agreement with the finding of Ross et al. [78], where the author confirms the dominance of hemispheric contribution in processing auditory stimuli in noisy environments.

The present study also investigated phase-sensitive binaural hearing using the time domain ERP peak analysis. The results revealed that the Na peak of the MLR for 500 Hz stimuli, the N1 peak of the LLR for 500 Hz stimuli, and the P300 peak of the LLR for 250 Hz stimuli show a statistically significant difference between the antiphasic and the homophasic condition for subject-wise analysis, indicating the importance of phase differences in binaural hearing. It has been suggested that neural functioning in the thalamo-cortical level (bottom up) and neurocognitive functions (top down) are related to phase-sensitive stimuli masked in noise for binaural hearing [79,80]. Furthermore, our study highlights the importance of the N1 peak and the P300 peak of LLR in the analysis of binaural hearing and their potential use as measures of cortical processing of IPD [80,81]. The Na peak of the MLR, the N1 peak of the LLR, and the P300 peak of the LLR are important components in the analysis of the relevance of interaural phase differences (IPD) for binaural hearing. The peaks are believed to reflect the processing of IPD at different levels of the auditory system, starting from the midbrain regions to the auditory cortex, and finally, the attentional and working memory systems. The Na peak of MLR is believed to represent the processing of IPD in the midbrain regions, specifically in the superior olivary complex and lateral lemniscus [82]. This peak is sensitive to small differences in IPD, making it an important measure for studying spatial hearing in binaural hearing tasks [83,84]. The N1 peak of LLR reflects cortical processing of IPD, particularly in the auditory cortex. This peak may represent the neural processing of the differences in the timing of the sound wave between the two ears at a higher level of the auditory system. Reduced N1 peak amplitudes may suggest a possible deficit in cortical processing of IPD, which may contribute to difficulties in discriminating tones and non-speech sounds from noise [85]. In literature, the N1 peak has been identified as a physiological index of the ability to “tune in” one’s attention to a single sound source when there are several competing sources in a noisy environment, again referring to the ‘cocktail party effect’ [86,87]. Enhancement of the N1 component for tasks which require selective attention has also been described in the literature [46] and is in line with the current study’s findings. The P300 peak of LLR may reflect cognitive processing of IPD, particularly in attentional and working memory systems. Several studies have stated the importance of the P300 component in analysing binaural hearing for normal adults as well as in adults with central processing disorders [88,89]. Reduced P300 peak amplitudes may suggest a possible deficit in cognitive processing of IPD and impaired attentional and working memory functions.

The frequency domain analysis results suggest that the brain is capable of detecting and processing phase differences in binaural hearing, particularly in the high beta and low gamma frequency range, and that optimal binaural processing occurs for 500 Hz stimuli, based on the MLR results. Additionally, our results provide guidance on the selection of electrode locations for future binaural hearing studies. For time domain analysis, the Na peak of the MLR and N1 peak of the LLR for 500 Hz stimuli can be used as markers in objective studies of binaural hearing. The P300 peaks of the LLR for 250 Hz stimuli also may contribute to objective measures for binaural hearing.

## 5. Conclusions

In conclusion, the current study explored the role of interaural phase differences in binaural processing in noise and their neural correlates in the human brain. The results demonstrated significant differences between auditory evoked potentials (AEP) generated by antiphasic and homophasic stimuli in both time and frequency domains. These findings highlight the brain’s ability to detect and process interaural phase differences, crucial for sound source localization and other auditory processing aspects.

Frequency domain analysis revealed significant differences in the middle latency response (MLR) signals for 500 Hz stimuli, while both 250 Hz and 500 Hz stimuli showed significant differences in the late latency response (LLR) signals, particularly in the 20–25 Hz and 25–30 Hz frequency bands. This suggests optimal binaural processing at 500 Hz, specifically in the high beta–low gamma frequency range, known for sensory integration. Additionally, midline electrodes proved more effective for investigating binaural processing of pure tone stimuli, yielding significant differences in MLR signals, while both the midline and left electrodes showed significant differences in LLR signals.

Furthermore, time domain analysis identified the Na peak of the MLR and N1 peak of the LLR for 500 Hz stimuli as significant markers for responses to homophasic and antiphasic stimuli, with potential applications in objective studies of binaural hearing. The P300 peak of the LLR for 250 Hz stimuli also exhibited strong significance between responses to homophasic and antiphasic stimuli, suggesting it might be considered as an objective measure for binaural hearing.

Future research can expand on these findings to explore the clinical implications of binaural processing in hearing disorders and related conditions.

## 6. Limitations

It is important to note, however, that the present study has some limitations. For instance, the sample size was thirty-five, and the current study mainly focused on healthy young adults, so the results may not be generalizable to other populations. Additionally, the study used a limited set of auditory stimuli, so future research could explore the effects of different types of stimuli on binaural processing in more depth.

## Figures and Tables

**Figure 1 jcm-12-04487-f001:**
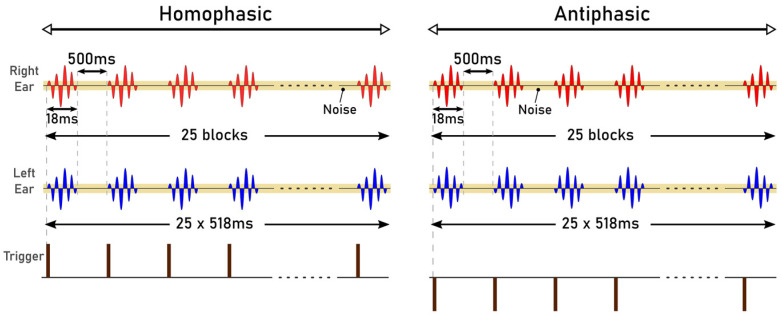
Auditory Stimuli of 18 ms.

**Figure 2 jcm-12-04487-f002:**
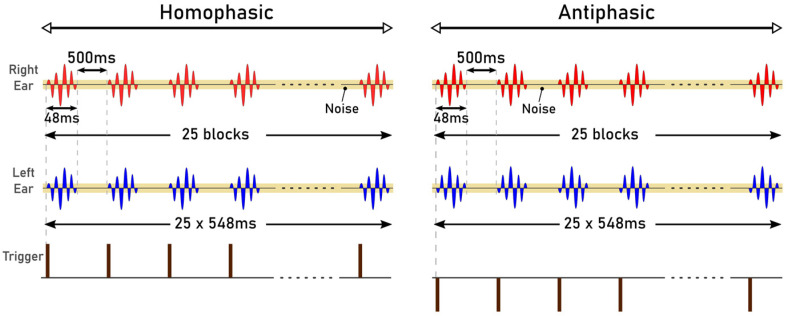
Auditory Stimuli of 48 ms.

**Figure 3 jcm-12-04487-f003:**
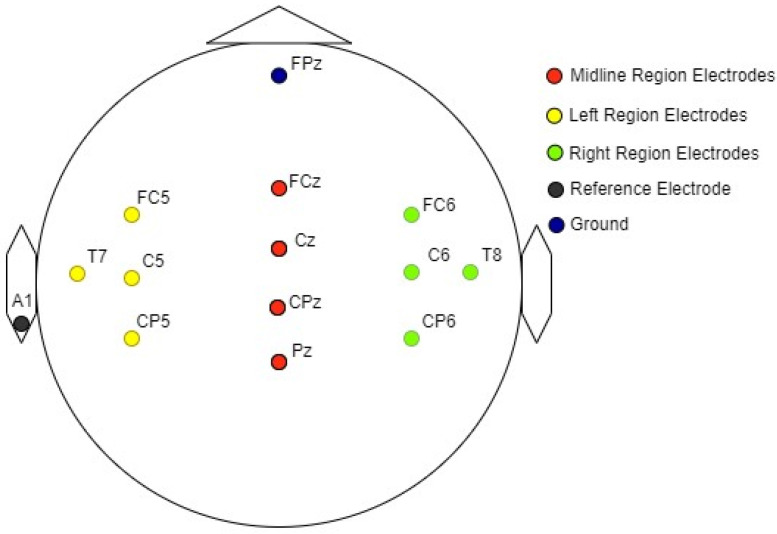
Graphical representation of the electrode arrangement for EEG measurements.

**Figure 4 jcm-12-04487-f004:**
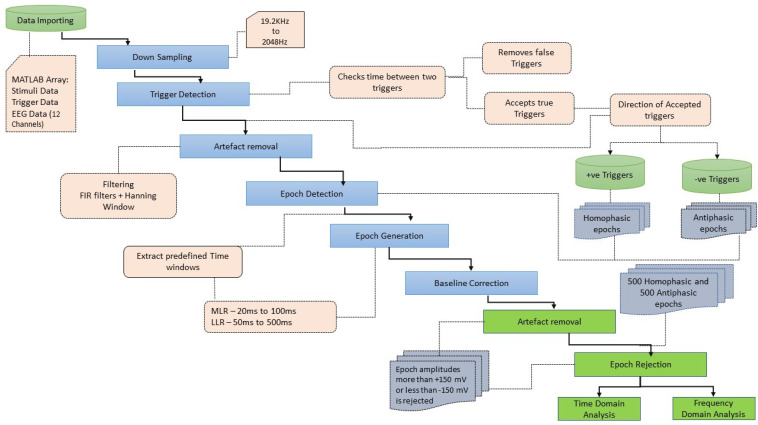
Workflow of data processing and analysis.

**Figure 5 jcm-12-04487-f005:**
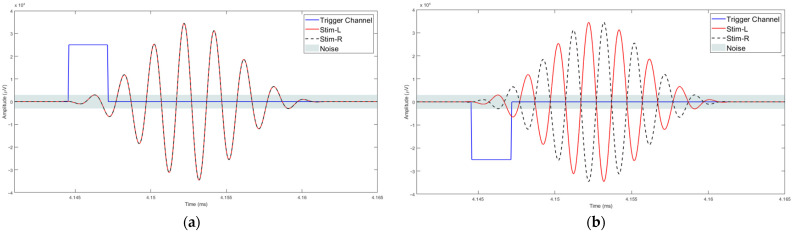
Trigger detection process: (**a**) Homophasic signal; (**b**) Antiphasic.

**Figure 6 jcm-12-04487-f006:**
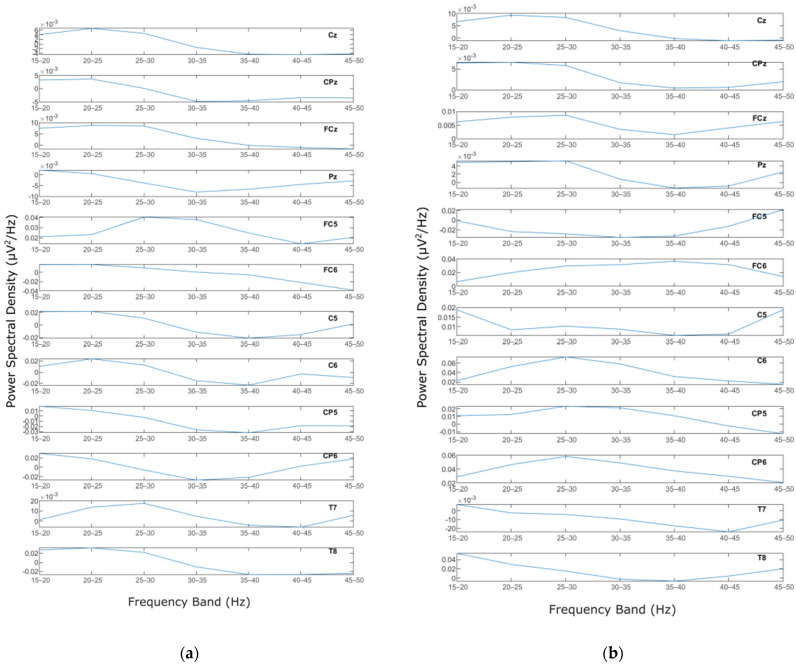
PSD for a representative subject: (**a**) Homophase; (**b**) Antiphase.

**Figure 7 jcm-12-04487-f007:**
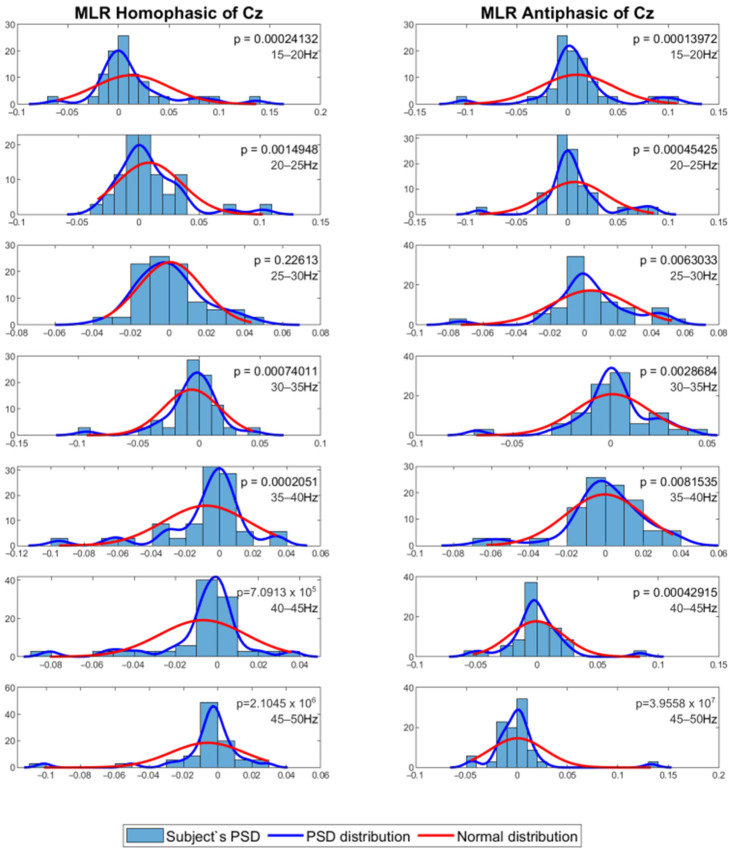
Normal distribution for seven different frequency bands for 500 Hz MLR Cz channel.

**Figure 8 jcm-12-04487-f008:**
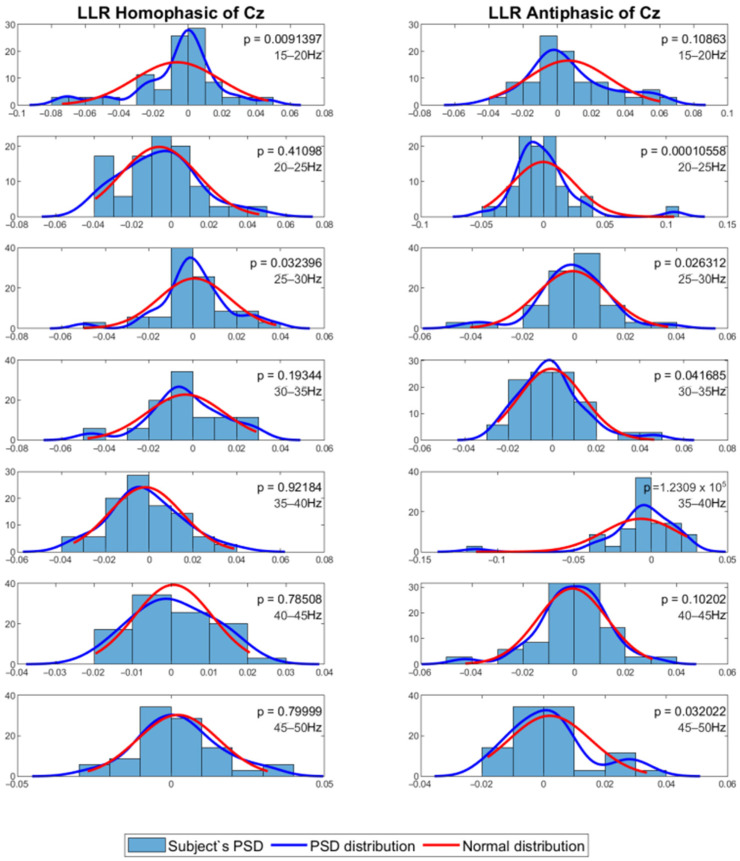
Normal distribution for seven different frequency bands for 500 Hz LLR Cz channel.

**Figure 9 jcm-12-04487-f009:**
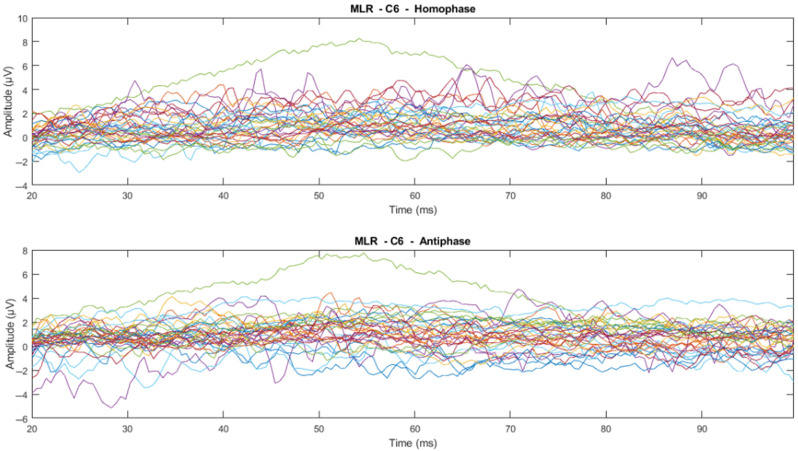
Channel-wise 500 Hz MLR ERP wave for all subjects at C6 electrode (Each line represents each subject).

**Figure 10 jcm-12-04487-f010:**
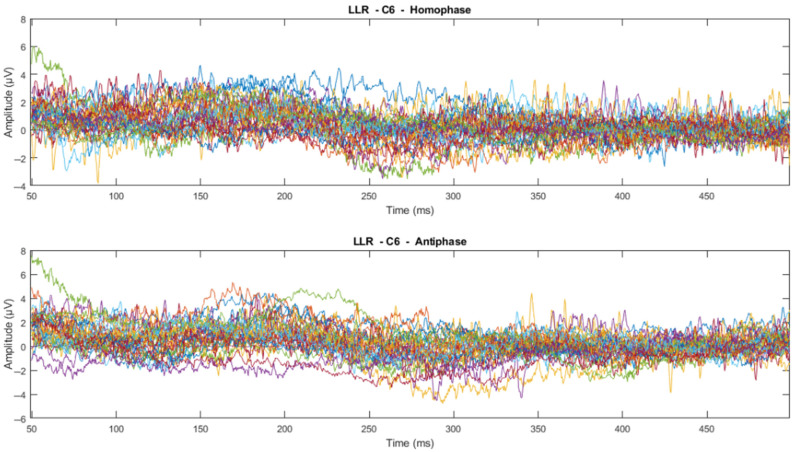
Channel-wise 500 Hz LLR ERP wave for all subjects at C6 electrode (Each line represents each subject).

**Figure 11 jcm-12-04487-f011:**
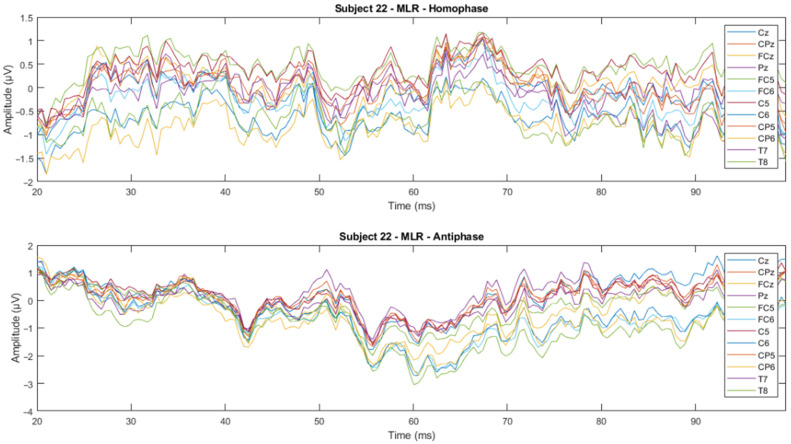
Subject-wise 500 Hz MLR ERP wave for all channels of Subject 22.

**Figure 12 jcm-12-04487-f012:**
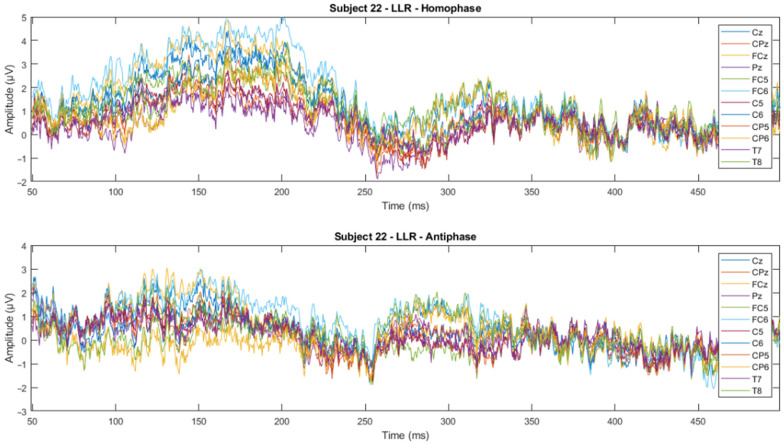
Subject-wise 500 Hz LLR ERP wave for all channels of Subject 22.

**Figure 13 jcm-12-04487-f013:**
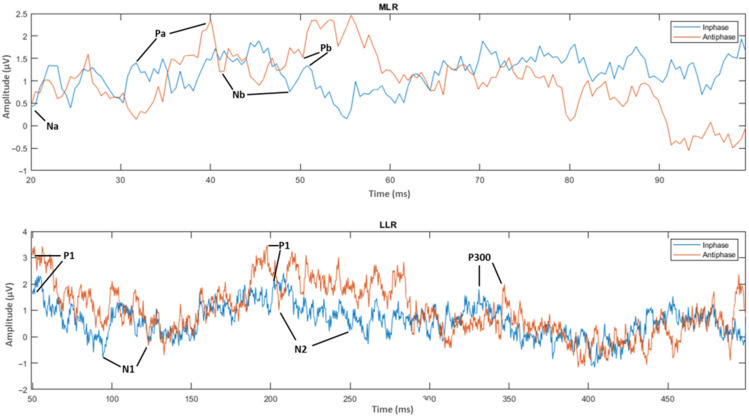
ERP wave peak components of homophase and antiphase for MLR and LLR.

**Figure 14 jcm-12-04487-f014:**
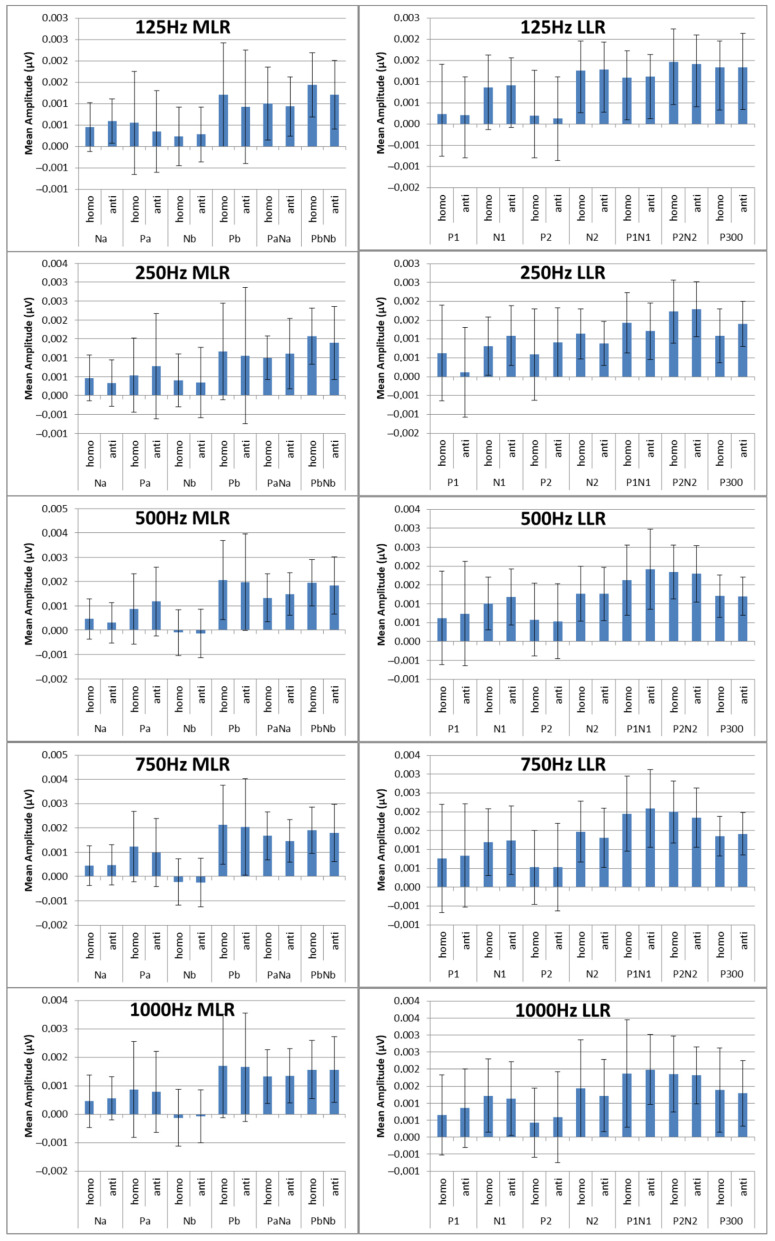
Mean peak amplitudes in (µV) with standard deviations for the ERP components for MLR.

**Table 1 jcm-12-04487-t001:** Wilcoxon signed-rank significance results for the MLR with125 Hz stimuli.

MLR_125 Hz								
	Frequency Bands	15–20 (Hz)	20–25 (Hz)	25–30 (Hz)	30–35 (Hz)	35–40 (Hz)	40–45 (Hz)	45–50 (Hz)
Channels								
Cz		0.342117	0.512361	0.869895	0.367667	0.287037	0.908719	0.869895
CPz		0.895749	0.646510	0.422219	0.325731	0.566462	0.973867	0.869895
FCz		0.359021	0.294516	0.973867	0.359021	0.163854	0.805924	0.908719
Pz		0.422219	0.394372	0.309864	0.359021	0.831382	0.960810	0.869895
FC5		0.376441	0.471105	0.481244	0.481244	0.694246	0.168869	0.104903
FC6		0.908719	0.481244	0.251572	0.127694	0.309864	0.219283	0.076903
C5		0.882807	0.831382	0.522962	0.768128	0.818630	0.882807	0.805924
C6		0.895749	0.682188	0.921714	0.412811	0.394372	0.302126	0.173998
CP5		0.342117	0.600186	0.768128	0.986932	0.768128	0.706381	0.947763
CP6		0.694246	0.960810	0.544496	0.682188	0.895749	0.793270	0.512361
T7		0.682188	0.646510	0.718592	0.394372	0.730875	0.706381	0.588844
T8		0.491501	0.743227	0.394372	0.512361	0.869895	0.441407	0.213200

**Table 2 jcm-12-04487-t002:** Wilcoxon signed-rank significance results for the MLR with 250 Hz stimuli.

MLR_250 Hz								
	Frequency Bands	15–20 (Hz)	20–25 (Hz)	25–30 (Hz)	30–35 (Hz)	35–40 (Hz)	40–45 (Hz)	45–50 (Hz)
Channels								
Cz		0.481244	0.706381	0.634790	0.251572	0.730875	0.857019	0.566462
CPz		0.730875	0.431751	0.279687	0.173998	0.611626	0.934730	0.658318
FCz		0.279687	0.279687	1.000000	0.755646	0.844180	0.869895	0.857019
Pz		0.658318	0.566462	0.287037	0.173998	0.471105	0.694246	0.566462
FC5		0.213200	0.085467	0.104903	0.144911	0.184604	0.385343	0.611626
FC6		0.805924	0.844180	0.694246	0.611626	0.973867	0.670212	0.973867
C5		0.522962	0.588844	0.471105	0.588844	0.805924	0.921714	0.793270
C6		0.921714	0.973867	0.973867	0.793270	0.718592	0.385343	0.238279
CP5		0.718592	0.844180	0.895749	0.501873	0.317733	0.309864	0.179243
CP6		0.718592	0.359021	0.634790	0.805924	0.908719	0.908719	0.706381
T7		0.140448	0.088487	0.066585	0.201400	0.566462	0.793270	0.706381
T8		0.325731	0.294516	0.431751	0.491501	0.302126	0.611626	0.973867

**Table 3 jcm-12-04487-t003:** Wilcoxon signed-rank significance results for the MLR with 500 Hz stimuli.

MLR_500 Hz								
	Frequency Bands	15–20 (Hz)	20–25 (Hz)	25–30 (Hz)	30–35 (Hz)	35–40 (Hz)	40–45 (Hz)	45–50 (Hz)
Channels								
Cz		0.085467	0.034609	0.020918	0.051281	0.158952	0.190084	0.385343
CPz		0.154161	0.016786	0.014667	0.149482	0.317733	0.294516	0.451185
FCz		0.127694	0.025910	0.030615	0.064192	0.225491	0.163854	0.207239
Pz		0.272465	0.030615	0.140448	0.682188	0.869895	0.367667	0.244862
FC5		0.461085	0.333859	0.287037	0.818630	0.869895	0.451185	0.190084
FC6		0.441407	0.908719	0.394372	0.184604	0.154161	0.244862	0.207239
C5		0.451185	0.294516	0.302126	0.359021	0.706381	0.566462	0.921714
C6		0.882807	0.818630	0.555426	0.522962	0.869895	0.743227	0.882807
CP5		0.066585	0.045689	0.149482	0.682188	0.818630	0.755646	0.960810
CP6		0.670212	0.577602	0.244862	0.670212	0.908719	1.000000	0.730875
T7		0.251572	0.272465	0.154161	0.163854	0.219283	0.149482	0.144911
T8		0.611626	0.566462	0.168869	0.131841	0.461085	0.882807	0.577602

**Table 4 jcm-12-04487-t004:** Wilcoxon signed-rank significance results for the MLR with750 Hz stimuli.

MLR_750 Hz								
	Frequency Bands	15–20 (Hz)	20–25 (Hz)	25–30 (Hz)	30–35 (Hz)	35–40 (Hz)	40–45 (Hz)	45–50 (Hz)
Channels								
Cz		0.658318	0.780670	0.934730	0.831382	0.646510	0.682188	0.646510
CPz		0.244862	0.422219	0.768128	0.818630	0.385343	0.403528	0.588844
FCz		0.076903	0.634790	0.577602	0.325731	0.238279	0.258408	0.367667
Pz		0.064192	0.333859	0.908719	0.768128	0.342117	0.325731	0.611626
FC5		0.028178	0.394372	0.265373	0.491501	0.960810	0.588844	0.908719
FC6		0.119704	0.173998	0.082530	0.272465	0.706381	0.522962	0.512361
C5		0.279687	0.491501	0.108459	0.025910	0.043943	0.385343	0.501873
C6		0.682188	0.350504	0.522962	0.947763	0.682188	0.805924	0.512361
CP5		0.908719	0.895749	0.857019	0.431751	0.461085	0.831382	0.857019
CP6		0.544496	0.119704	0.231822	0.882807	0.501873	0.123648	0.131841
T7		0.094786	0.555426	0.471105	0.302126	0.294516	0.512361	0.317733
T8		0.441407	0.780670	0.658318	0.755646	0.555426	0.049357	0.085467

**Table 5 jcm-12-04487-t005:** Wilcoxon signed-rank significance results for the MLR with 1000 Hz stimuli.

MLR_1000 Hz								
	Frequency Bands	15–20 (Hz)	20–25 (Hz)	25–30 (Hz)	30–35 (Hz)	35–40 (Hz)	40–45 (Hz)	45–50 (Hz)
Channels								
Cz		0.294516	0.376441	0.094786	0.055319	0.104903	0.184604	0.190084
CPz		0.317733	0.611626	0.115858	0.123648	0.376441	0.333859	0.219283
FCz		0.219283	0.350504	0.309864	0.201400	0.251572	0.342117	0.461085
Pz		0.244862	0.718592	0.544496	0.670212	0.793270	0.265373	0.140448
FC5		0.670212	0.544496	0.818630	0.522962	0.611626	0.611626	0.431751
FC6		0.947763	0.566462	0.333859	0.082530	0.064192	0.076903	0.207239
C5		0.491501	0.251572	0.694246	0.805924	0.895749	0.163854	0.005929
C6		0.895749	0.694246	0.422219	0.195682	0.461085	0.201400	0.131841
CP5		0.986932	0.869895	0.670212	0.755646	0.818630	0.127694	0.074208
CP6		0.755646	0.611626	0.279687	0.251572	0.682188	0.115858	0.047493
T7		0.219283	0.244862	0.960810	0.471105	0.294516	0.294516	0.258408
T8		0.302126	0.768128	0.986932	0.895749	0.934730	0.471105	0.225491

**Table 6 jcm-12-04487-t006:** Wilcoxon signed-rank significance results for the LLR with 125 Hz stimuli.

LLR_125 Hz								
	Frequency Bands	15–20 (Hz)	20–25 (Hz)	25–30 (Hz)	30–35 (Hz)	35–40 (Hz)	40–45 (Hz)	45–50 (Hz)
Channels								
Cz		0.895749	0.960810	0.730875	0.831382	0.385343	0.294516	0.512361
CPz		0.646510	0.818630	0.805924	0.831382	0.566462	0.422219	0.623162
FCz		0.588844	0.844180	0.325731	0.934730	0.317733	0.350504	0.566462
Pz		0.908719	0.973867	0.805924	0.658318	0.566462	0.385343	0.646510
FC5		0.718592	0.947763	0.577602	0.061871	0.333859	0.272465	0.844180
FC6		0.588844	0.973867	0.986932	0.251572	0.094786	0.140448	0.947763
C5		0.359021	0.973867	0.986932	0.094786	0.119704	0.076903	0.522962
C6		0.471105	0.844180	0.441407	0.544496	0.461085	0.422219	0.818630
CP5		0.934730	0.831382	0.780670	0.244862	0.158952	0.244862	0.350504
CP6		0.768128	0.793270	0.682188	0.844180	0.287037	0.367667	0.600186
T7		0.844180	0.168869	0.611626	0.818630	0.272465	0.566462	0.670212
T8		0.646510	0.768128	0.394372	0.947763	0.394372	0.213200	0.238279

**Table 7 jcm-12-04487-t007:** Wilcoxon signed-rank significance results for the LLR with 250 Hz stimuli.

LLR_250 Hz								
	Frequency Bands	15–20 (Hz)	20–25 (Hz)	25–30 (Hz)	30–35 (Hz)	35–40 (Hz)	40–45 (Hz)	45–50 (Hz)
Channels								
Cz		0.555426	0.213200	0.047493	0.768128	0.265373	0.755646	0.718592
CPz		0.844180	0.441407	0.163854	0.512361	0.359021	0.385343	0.844180
FCz		0.611626	0.190084	0.040620	0.755646	0.173998	0.882807	0.857019
Pz		0.694246	0.960810	0.294516	0.718592	0.670212	0.190084	0.623162
FC5		0.163854	0.085467	0.098068	0.921714	0.158952	0.921714	0.461085
FC6		0.302126	0.258408	0.119704	0.431751	0.793270	0.730875	0.461085
C5		0.213200	0.158952	0.016052	0.921714	0.533674	0.718592	0.986932
C6		0.461085	0.367667	0.512361	0.287037	0.588844	0.309864	0.163854
CP5		0.441407	0.272465	0.279687	0.623162	0.566462	0.818630	0.706381
CP6		0.706381	0.947763	0.857019	0.947763	0.921714	0.403528	0.173998
T7		0.112111	0.287037	0.119704	0.623162	0.986932	0.780670	0.279687
T8		0.600186	0.831382	0.869895	0.431751	0.973867	0.577602	0.566462

**Table 8 jcm-12-04487-t008:** Wilcoxon signed-rank significance results for the LLR with 500 Hz stimuli.

LLR_500 Hz								
	Frequency Bands	15–20 (Hz)	20–25 (Hz)	25–30 (Hz)	30–35 (Hz)	35–40 (Hz)	40–45 (Hz)	45–50 (Hz)
Channels								
Cz		0.512361	0.213200	0.064192	0.481244	0.831382	0.588844	0.441407
CPz		0.844180	0.287037	0.053268	0.422219	0.921714	0.646510	0.718592
FCz		0.706381	0.317733	0.016786	0.658318	0.793270	0.544496	0.441407
Pz		0.694246	0.251572	0.022804	0.302126	0.947763	0.611626	0.934730
FC5		0.195682	0.317733	0.154161	0.043943	0.555426	0.658318	0.104903
FC6		0.101440	0.049357	0.127694	0.471105	0.646510	0.805924	0.522962
C5		0.225491	0.085467	0.491501	0.805924	0.403528	0.451185	0.098068
C6		0.201400	0.140448	0.040620	0.112111	0.934730	0.960810	0.006881
CP5		0.294516	0.127694	0.016052	0.074208	0.882807	0.294516	0.195682
CP6		0.231822	0.512361	0.076903	0.670212	0.805924	0.743227	0.057436
T7		0.127694	0.011124	0.022804	0.079676	0.934730	0.555426	0.279687
T8		0.091593	0.079676	0.012788	0.074208	0.184604	0.461085	0.682188

**Table 9 jcm-12-04487-t009:** Wilcoxon signed-rank significance results for the LLR with750 Hz stimuli.

LLR_750 Hz								
	Frequency Bands	15–20 (Hz)	20–25 (Hz)	25–30 (Hz)	30–35 (Hz)	35–40 (Hz)	40–45 (Hz)	45–50 (Hz)
Channels								
Cz		0.195682	0.173998	0.818630	0.818630	0.646510	0.805924	0.471105
CPz		0.064192	0.251572	0.844180	0.544496	0.385343	0.706381	0.385343
FCz		0.461085	0.272465	0.265373	0.600186	0.600186	0.818630	0.670212
Pz		0.342117	0.037512	0.682188	0.238279	0.213200	0.658318	0.921714
FC5		0.265373	0.882807	0.844180	0.481244	0.394372	0.805924	0.168869
FC6		0.921714	0.611626	0.577602	0.294516	0.600186	0.882807	0.094786
C5		0.818630	0.385343	0.294516	0.359021	0.333859	0.376441	0.921714
C6		0.219283	0.588844	0.857019	0.422219	0.755646	0.325731	0.025910
CP5		0.544496	0.238279	0.471105	0.793270	0.743227	0.317733	0.491501
CP6		0.158952	0.333859	0.908719	0.403528	0.986932	0.522962	0.112111
T7		0.960810	0.034609	0.030615	0.555426	0.634790	0.367667	0.646510
T8		0.333859	0.441407	0.805924	0.082530	0.127694	0.238279	0.057436

**Table 10 jcm-12-04487-t010:** Wilcoxon signed-rank significance results for the LLR with1000 Hz stimuli.

LLR_1000 Hz								
	Frequency Bands	15–20 (Hz)	20–25 (Hz)	25–30 (Hz)	30–35 (Hz)	35–40 (Hz)	40–45 (Hz)	45–50 (Hz)
Channels								
Cz		0.934730	1.000000	0.986932	0.219283	0.238279	0.251572	0.533674
CPz		0.611626	0.522962	0.743227	0.213200	0.213200	0.258408	0.367667
FCz		0.857019	0.768128	0.730875	0.350504	0.098068	0.140448	0.501873
Pz		0.522962	0.451185	0.600186	0.522962	0.376441	0.244862	0.367667
FC5		0.623162	0.857019	0.818630	0.251572	0.287037	0.140448	0.123648
FC6		0.451185	0.768128	0.611626	0.403528	0.317733	0.020918	0.195682
C5		0.168869	0.168869	0.588844	0.908719	0.623162	0.158952	0.706381
C6		0.755646	0.780670	0.219283	0.244862	0.127694	0.108459	0.682188
CP5		0.934730	0.973867	0.780670	0.682188	0.857019	0.947763	0.805924
CP6		0.805924	0.512361	0.207239	0.025910	0.055319	0.179243	0.179243
T7		0.112111	0.611626	0.376441	0.793270	0.730875	0.422219	0.908719
T8		0.533674	0.793270	0.359021	0.207239	0.055319	0.588844	0.555426

**Table 11 jcm-12-04487-t011:** Variable categories extracted from the ERPs.

MLR	LLR
Na Peak	N1 Peak
Nb Peak	N2 Peak
Pa Peak	P1 Peak
Pb Peak	P2 Peak
PaNa	P1N1
PbNb	P2N2
	P300

**Table 12 jcm-12-04487-t012:** Significance test for absolute peaks between antiphase and homophase for MLR.

Two-Sample T-Test_MLR						
Frequency	Na_Peak	Nb_Peak	PaNa_Peak	Pa_Peak	PbNb_Peak	Pb_Peak
125	0.249583	0.557758	0.09326368	0.550172	0.43695471	0.190337
250	0.85548	0.433864	0.06306216	0.439261	0.38729981	0.755598
500	0.048271	0.678389	0.53669633	0.804749	0.81264522	0.912804
750	0.597208	0.34431	0.94425832	0.378184	0.87791895	0.779411
1000	0.761097	0.955758	0.16387057	0.731612	0.60889488	0.767744

**Table 13 jcm-12-04487-t013:** Significance test for absolute peaks between antiphase and homophase for LLR.

Two-Sample T-Test_LLR							
Frequency	N1_Peak	N2_Peak	P1N1_Peak	P1_Peak	P2N2_Peak	P2_Peak	P300_Peak
125	0.66272	0.89587	0.22803	0.78507	0.60968	0.52700	0.55595
250	0.13246	0.10164	0.43950	0.17504	0.89817	0.81076	0.03309
500	0.02776	0.86421	0.60571	0.46280	0.78454	0.91343	0.54314
750	0.66033	0.34891	0.57140	0.66904	0.66811	0.47989	0.56215
1000	0.80181	0.60165	0.45661	0.31409	0.46152	0.44283	0.78263

## Data Availability

Not applicable.

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
