# Peer review of "Frequency and Time Domain Analysis of EEG Based Auditory Evoked Potentials to Detect Binaural Hearing in Noise"

_jcm, 2023, doi:10.3390/jcm12134487_

Round 1

Reviewer 1 Report

Frequency and Time Domain Analysis of EEG based Auditory 2 Evoked Potentials to Detect Binaural Hearing in Noise

Minor revisions are required:

Abstract

The authors do not mention the characteristics of the sample.

Introduction: it is excessively long and deserves an adequate summary to focus attention on the objective of the study

Line 39 A list of causes of hearing loss is not essential for the purpose of the study

Line 49 Why do you only mention conductive hearing loss in this phrase?

Line 118-line 123 Is a repetition.

Line 130-149 this part deserves a summary as there are many repetitions

From line 151 to 200 The length of this section relating to the technical characteristics of the recording distracts the reader and loses interest. specify these characteristics in the methods and summarize a lot in the introduction.

Line 202 is a repetition

Line 205 What is BMLD? you have not used this acronym before in the text; explain.

Line 209 Do not anticipate the results in the introduction but better list the objectives of the study

Materials and Methods

Participants: a detailed list of inclusion and exclusion criteria is missing which makes the study not reproducible

Specify the age range of the subjects included, the mean age and the standard deviation, the male/female ratio.

The hearing threshold of 20 dB HL in all patients seems contrived; calculate the mean PTA and the standard deviation

Auditory Stimuli

How were auditory stimuli sent? headphones? Specify

Results and analysis

I personally do not appreciate the union that has been made between the methods of analysis and the results. I leave the decision to the editor but in my opinion separating the sections facilitates the reading of the article and makes the results clearer to the reader.

Discussion and conclusions

No comment to make; The text turns out to be sufficiently explanatory, concise and pleasant to read compared to the other parts that are more confusing.

Author Response

Thank you for the valuable time you have taken for reviewing our work. We have answered all the concerns you have raised. Please see the attachment for our responses and actions towards the points.

Reviewer#1, Concern # 1:
Concerns:
Abstract: The authors do not mention the characteristics of the sample.
Author response & action: First of all, thank you very much for time that you have given for the thorough review of our paper. We have now updated the abstract of the manuscript (in line 15 of page 1) with the characteristics of the sample. We highly appreciate this valuable observation.

Reviewer#1, Concern # 2:
Concerns:
Introduction: Line 39 A list of causes of hearing loss is not essential for the purpose of the study.
Author response & action: Thank you for pointing out this. We appreciate the suggestion and those lines (line 40-46) have now been removed to make the introduction as close as to the purpose of study.

Reviewer#1, Concern # 3:
Concerns:
Introduction: Line 49 Why do you only mention conductive hearing loss in this phrase?
Author response & action: Thank you for letting us know this point. Both types of hearing loss with references is now described in lines 50-52 of page 2.

Reviewer#1, Concern # 4:
Concerns:
Introduction: Line 118-line 123 Is a repetition.
Author response & action: Thank you for the observation and sorry for making repetitions. We have now updated the manuscript by removing the repeated lines.

Reviewer#1, Concern # 5:
Concerns:
Introduction: Line 130-149 this part deserves a summary as there are many repetitions.
Author response & action: Thank you for your valuable suggestion. The summarised lines are now added to the manuscript in 132-137 and 143 to 149.

Reviewer#1, Concern # 6:
Concerns:
Introduction: From line 151 to 200 The length of this section relating to the technical characteristics of the recording distracts the reader and loses interest. specify these characteristics in the methods and summarize a lot in the introduction.
Author response & action: Thank you for your valuable observation and suggestion. We have now updated the manuscript by removing the technical characteristics from introduction section. The changes are done in page 4 of the manuscript.

Reviewer#1, Concern # 7:
Concerns:
Introduction: Line 202 is a repetition.
Author response & action: Thank you for the observation and sorry for making repetitions. The updated manuscript has now removed the repeated line 202.

Reviewer#1, Concern # 8:
Concerns:
Introduction: Line 205 What is BMLD? you have not used this acronym before in the text; explain.
Author response & action: Thank you for your observation. It was our mistake, and we admit that. We have now explained the abbreviation ‘BMLD’ in the line 233.

Reviewer#1, Concern # 9:
Concerns:
Introduction: Line 209 Do not anticipate the results in the introduction but better list the objectives of the study.
Author response & action: Thank you for your valuable comment. We have now updated the introduction section as suggested and is seen lines 228-240 of page 5.

Reviewer#1, Concern # 10:
Concerns:
Materials and Methods: Participants: a detailed list of inclusion and exclusion criteria is missing which makes the study not reproducible. Specify the age range of the subjects included, the mean age and the standard deviation, the male/female ratio. The hearing threshold of 20 dB HL in all patients seems
contrived; calculate the mean PTA and the standard deviation.
Author response & action: Thank you for the valuable comments on the participant’s section of the manuscript. We have now described the required information in detail as suggested. The updation can be viewed in lines 259 - 275.

Reviewer#1, Concern # 11:
Concerns:
Materials and Methods: Auditory Stimuli: How were auditory stimuli sent? headphones? Specify

Author response & action: Thank you for the observation. We have now updated the manuscript with the relevant information on the delivery of auditory stimuli in lines 297-299.

Reviewer#1, Concern # 12:
Concerns:
Results and Analysis: I personally do not appreciate the union that has been made between the methods of analysis and the results. I leave the decision to the editor but in my opinion separating the sections facilitates the reading of the article and makes the results clearer to the reader.

Author response & action: Thank you for your valuable comment. We are analysing the results one after another in a continuous manner and with respect to the previous results obtained. Hence, it’s better for us to write both sections together to convey our results and findings to the readers. Also, the separation of section as suggested can affect the entire structure of the current manuscript. We highly appreciate your suggestion and in our future works we will be taking your suggestion on writing into consideration.

Reviewer#1, Concern # 13:
Concerns:
Discussions and conclusions: No comment to make; The text turns out to be sufficiently explanatory, concise and pleasant to read compared to the other parts that are more confusing.
Author response & action: Thank you for the valuable comments on the Discussions and conclusion section.

Editor Comment while submitted:
Concerns:
Missing of details of author contributions, funding, ethics approval etc at the end of the manuscript.
Author response & action: Thank you for the valuable comments and now the manuscript is updated with required information in lines 667 - 684.

Reviewer 2 Report

The work is of great practical and scientific interest. The topic is very relevant, especially considering the annually growing number of patients with  hearing loss.

This study is unique in that it focuses on phase changes and uses stimuli with frequency and noise parameters. The findings may improve our understanding of binaural processing in the human brain and lead to applications in the development of new objective hearing tests in the future.

For the first time such an analysis was carried out between  by antiphasic and homophasic stimuli  in binaural processing in the human brain by auditory evoked potentials (AEPs) and demonstrates  brain's ability to detect and process interaural phase differences, crucial for sound source localization and other auditory processing aspects.

The purpose of the study is described clearly.

The rational for selecting that particular statistic, and which variables were entered into the statistic are described.

statistical methods of comparison were chosen very competently.

Statistical results are presented in a Tables and Figures.

Long introduction. However, this is justified due to the explanation of many terms and concepts that may be new to the reader. The introduction discusses the main issues related to the topic, with reference to contemporary sources. The results are presented and demonstrated in an understandable scientific language using comparative and correlation analysis. Results are structured around the Research Questions.

The title of the article matches the content. The purpose and objectives of the work are fully realized.

Based on the results of this study, the authors came up with important findings with the determination of reliable frequency ranges and localization of recording electrodes for registration middle latency responses  late latency responses.

Discussion and conclusions follow logically from the results of the study and are fully consistent with the purpose of the study.

The main findings as related to the overall purpose of the study are discussed and explained in detail.

Conclusions is directly related to the data that was collected and analyzed.

Notes:

1.       Figure â„– 4 is not readable. use a picture with a higher resolution.

2.       Figure â„– 5 is not readable. use a picture with a higher resolution.

3.       Figure â„– 6 is not readable. use a picture with a higher resolution.

4.       Figure â„– 14 is not legible. use a picture with a higher resolution.

5.       The list of references is not formatted according to the rules of the journal

6.       A lot of numbers and formulas are not interesting to read and difficult to concentrate.

7.       Table â„– 1. Indicate what the numbers are in the second row of the table. If this is age, then make the appropriate changes in the description of the table.

8.       Table â„– 2. Indicate what the numbers are in the second row of the table. If this is age, then make the appropriate changes in the description of the table.

9.       Table â„– 3. Indicate what the numbers are in the second row of the table. If this is age, then make the appropriate changes in the description of the table.

10.   Table â„– 4. Indicate what the numbers are in the second row of the table. If this is age, then make the appropriate changes in the description of the table.

11.   Table â„– 5. Indicate what the numbers are in the second row of the table. If this is age, then make the appropriate changes in the description of the table.

12.   Table â„– 6. Indicate what the numbers are in the second row of the table. If this is age, then make the appropriate changes in the description of the table.

13.   Table â„– 7. Indicate what the numbers are in the second row of the table. If this is age, then make the appropriate changes in the description of the table.

14.   Table â„– 8. Indicate what the numbers are in the second row of the table. If this is age, then make the appropriate changes in the description of the table.

15.   Table â„– 9. Indicate what the numbers are in the second row of the table. If this is age, then make the appropriate changes in the description of the table.

16.   Table â„– 10. Indicate what the numbers are in the second row of the table. If this is age, then make the appropriate changes in the description of the table.

17.   Make sure that abbreviations of terms and their descriptions are at the beginning of the text

Author Response

Thank you for the valuable time you have taken for reviewing our work. We have answered all the concerns you have raised. Please see the attachment for our responses and actions towards the points.

Reviewer#2, Concern # 1:
Concerns: Figure â„– 4 is not readable. use a picture with a higher resolution.
Author response & action: First of all, thank you for the time and effort you have taken to do a thorough review of our paper. We have now replaced Figure No 4 with a higher resolution image.

Reviewer#2, Concern # 2:
Concerns: Figure â„– 5 is not readable. use a picture with a higher resolution.
Author response & action: Thank you for your concern. We have now replaced Figure No 5 with a higher resolution image.

Reviewer#2, Concern # 3:
Concerns: Figure â„– 6 is not readable. use a picture with a higher resolution.
Author response & action: Thank you for the observation. We have now replaced Figure No 6 with a higher resolution image.

Reviewer#2, Concern # 4:
Concerns: Figure â„– 14 is not legible. use a picture with a higher resolution.
Author response & action: Thank you for your concern. We have now replaced Figure No 14 with a higher resolution image.

Reviewer#2, Concern # 5:
Concerns: The list of references is not formatted according to the rules of the journal.
Author response & action: Thank you for the observation. We have now formatted all the references according to the journal rules.

Reviewer#2, Concern # 6:
Concerns: A lot of numbers and formulas are not interesting to read and difficult to concentrate.
Author response & action: Thank you for pointing out this. However, it was essential to include all the current tables, numbers, and formulas even we know they are not much interesting to read and difficult to concentrate. So, without these all its not possible to explain the current findings of the study. We highly appreciate this valuable observation.

Reviewer#2, Concern # 7:
Concerns: Table â„– 1. Indicate what the numbers are in the second row of the table. If this is age, then make the appropriate changes in the description of the table.
Table â„– 2. Indicate what the numbers are in the second row of the table. If this is age, then make the appropriate changes in the description of the table.
Table â„– 3. Indicate what the numbers are in the second row of the table. If this is age, then make the appropriate changes in the description of the table.
Table â„– 4. Indicate what the numbers are in the second row of the table. If this is age, then make the appropriate changes in the description of the table.
Table â„– 5. Indicate what the numbers are in the second row of the table. If this is age, then make the appropriate changes in the description of the table.
Table â„– 6. Indicate what the numbers are in the second row of the table. If this is age, then make the appropriate changes in the description of the table.
Table â„– 7. Indicate what the numbers are in the second row of the table. If this is age, then make the appropriate changes in the description of the table.
Table â„– 8. Indicate what the numbers are in the second row of the table. If this is age, then make the appropriate changes in the description of the table.
Table â„– 9. Indicate what the numbers are in the second row of the table. If this is age, then make the appropriate changes in the description of the table.
Table â„– 10. Indicate what the numbers are in the second row of the table. If this is age, then make the appropriate changes in the description of the table.
Author response & action: Thank you for your observation and it was our mistake of not providing the title separate for the second row of tables from 1 to 10. They are the frequency bands under analysis. We admit our mistake and we have now given the separate titles for all the tables from 1 to 10.

Reviewer#2, Concern # 8:
Concerns: Make sure that abbreviations of terms and their descriptions are at the beginning of the text.
Author response & action: Thank you for the valuable concern. We have now double checked the issue and now have made sure that all the abbreviations of terms and their descriptions are at the beginning of the text.

Editor Comment while submitted:
Concerns:
Missing of details of author contributions, funding, ethics approval etc at the end of the manuscript. 

Author response & action: Thank you for the valuable comments and now the manuscript is updated with required information in lines 667 - 684.
